# WaveBound: Dynamic Error Bounds for Stable Time Series Forecasting

**Youngin Cho***    **Daejin Kim***    **Dongmin Kim**    **Mohammad Azam Khan**    **Jaegul Choo**
KAIST AI
{choyi0521,kiddj,tommy.dm.kim,azamkhan,jchoo}@kaist.ac.kr

## Abstract

Time series forecasting has become a critical task due to its high practicality in real-world applications such as traffic, energy consumption, economics and finance, and disease analysis. Recent deep-learning-based approaches have shown remarkable success in time series forecasting. Nonetheless, due to the dynamics of time series data, deep networks still suffer from unstable training and overfitting. Inconsistent patterns appearing in real-world data lead the model to be biased to a particular pattern, thus limiting the generalization. In this work, we introduce the dynamic error bounds on training loss to address the overfitting issue in time series forecasting. Consequently, we propose a regularization method called *WaveBound* which estimates the adequate error bounds of training loss for each time step and feature at each iteration. By allowing the model to focus less on unpredictable data, WaveBound stabilizes the training process, thus significantly improving generalization. With the extensive experiments, we show that *WaveBound* consistently improves upon the existing models in large margins, including the state-of-the-art model.

## 1   Introduction

Time series forecasting has gained a lot of attention due to its high practicality in real-world applications such as traffic [1], energy consumption [2], economics and finance [3], and disease analysis [4]. Recent deep-learning-based approaches, particularly transformer-based methods [5, 6, 7, 8, 9], have shown remarkable success in time series forecasting. Nevertheless, inconsistent patterns and unpredictable behaviors in real data enforce the models to fit in patterns, even for the cases of *unpredictable* incident, and induce the unstable training. In unpredictable cases, the model does not neglect them in training, but rather receives a huge penalty (*i.e.*, training loss). Ideally, small magnitudes of training loss should be presented for unpredictable patterns. This implies the need for proper regularization of the forecasting models in time series forecasting.

Recently, Ishida *et al.* [10] claimed that zero training loss introduces a high bias in training, hence leading to an overconfident model and a decrease in generalization. To remedy this issue, they propose a simple regularization called *flooding* and explicitly prevent the training loss from decreasing below a small constant threshold called the *flood level*. In this work, we also focus on the drawbacks of *zero training loss* in time series forecasting. In time series forecasting, the model is enforced to fit to an inevitably appearing unpredictable pattern which mostly generates a tremendous error. However, the original flooding is not applicable to time series forecasting mainly due to the following two main reasons. (i) Unlike image classification, time series forecasting requires the vector output of the size of the prediction length times the number of features. In this case, the original flooding considers the average training loss without dealing with each time step and feature individually. (ii) In time series

---

*Equal contribution

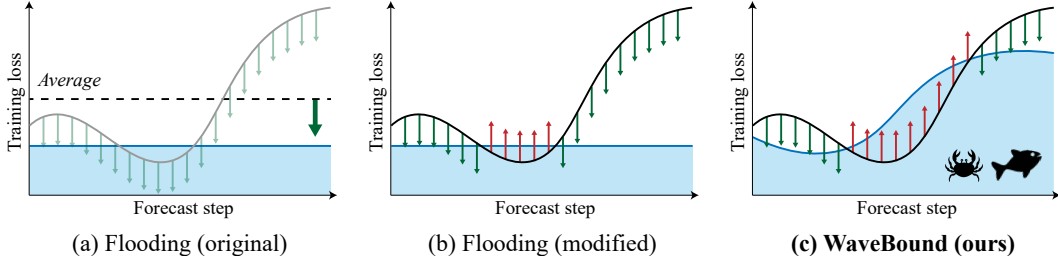

Figure 1: The conceptual examples for different methods. (a) The original flooding provides the lower bound of the average loss, rather than considering each time step and feature individually. (b) Even if the lower bounds of training loss are provided for each time step and feature, the bound of constant value cannot reflect the nature of time series forecasting. (c) Our proposed WaveBound method provides the lower bound of the training loss for each time step and feature. This lower bound is *dynamically adjusted* to give a tighter error bound during the training process.

data, error bounds should be dynamically changed for different patterns. Intuitively, a higher error should be tolerated for unpredictable patterns.

To properly address the overfitting issue in time series forecasting, the difficulty of prediction, *i.e.*, how unpredictable the current label is, should be measured in the training procedure. To this end, we introduce the *target network* updated with an exponential moving average of the original network, *i.e.*, *source network*. At each iteration, the target network can guide a reasonable level of training loss to the source network — the larger the error of the target network, the more unpredictable the pattern. In current studies, a slow-moving average target network is commonly used to produce stable targets in the self-supervised setting [11, 12]. By using the training loss of the target network for our lower bound, we derive a novel regularization method called *WaveBound* which faithfully estimates the error bounds for each time step and feature. By dynamically adjusting the error bounds, our regularization prevents the model from overly fitting to a certain pattern and further improves generalization. Figure 1 shows the conceptual difference between the original flooding and our WaveBound method. The originally proposed flooding determines the direction of the update step for all points by comparing the average loss and its flood level. In contrast, WaveBound individually decides the direction of the update step for each point by using the dynamic error bound of the training loss. The difference between these methods is further discussed in Section 3. Our main contributions are threefold:

- We propose a simple yet effective regularization method called *WaveBound* that dynamically provides the error bounds of training loss in time series forecasting.
- We show that our proposed regularization method consistently improves upon the existing state-of-the-art time series forecasting model on six real-world benchmarks.
- By conducting extensive experiments, we verify the significance of adjusting the error bounds for each time step, feature, and pattern, thus addressing the overfitting issue in time series forecasting.

## 2 Preliminary

### 2.1 Time Series Forecasting

We consider the rolling forecasting setting with a fixed window size [5, 6, 7]. The aim of time series forecasting is to learn a forecaster $g : \mathbb{R}^{L \times K} \to \mathbb{R}^{M \times K}$ which predicts the future series $y^t = \{z_{t+1}, z_{t+2}, ..., z_{t+M} : z_i \in \mathbb{R}^K\}$ given the past series $x^t = \{z_{t-L+1}, z_{t-L+2}, ..., z_t : z_i \in \mathbb{R}^K\}$ at time $t$ where $K$ is the feature dimension and $L$ and $M$ are the input length and output length, respectively. We mainly address the error bounding in the multivariate regression problem where the input series $x$ and output series $y$ jointly come from the underlying density $p(x, y)$. For a given loss function $\ell$, the risk of $g$ is $R(g) := \mathbb{E}_{(x,y) \sim p(x,y)} [\ell(g(x), y)]$. Since we cannot directly access the distribution $p$, we instead minimize its empirical version $\hat{R}(g) := \frac{1}{N} \sum_{i=1}^{N} \ell(g(x_i), y_i)$ using training data $\mathcal{X} := \{(x_i, y_i)\}_{i=1}^{N}$. In the analysis, we assume that the errors are independent and identically distributed. We mainly consider using the mean squared error (MSE) loss, which is widely used as an

objective function in recent time series forecasting models [5, 6, 7]. Then, the risk can be rewritten as the sum of the risk at each prediction step and feature:

$$R(g) = \mathbb{E}_{(u,v)\sim p(u,v)}\left[\frac{1}{MK}||g(u) - v||^2\right] = \frac{1}{MK}\sum_{j,k} R_{jk}(g),$$

$$\hat{R}(g) = \frac{1}{NMK}\sum_{i=1}^{N}||g(x_i) - y_i||^2 = \frac{1}{MK}\sum_{j,k}\hat{R}_{jk}(g),$$

(1)

where $R_{jk}(g) := \mathbb{E}_{(u,v)\sim p(u,v)}\left[||g_{jk}(u) - v_{jk}||^2\right]$ and $\hat{R}_{jk}(g) := \frac{1}{N}\sum_{i=1}^{N}||g_{jk}(x_i) - (y_i)_{jk}||^2$.

## 2.2 Flooding

To address the overfitting problem, Ishida *et al.* [10] suggested *flooding*, which restricts the training loss to stay above a certain constant. Given the empirical risk $\hat{R}$ and the manually searched lower bound $b$, called the *flood level*, we instead minimize the flooded empirical risk, which is defined as

$$\hat{R}^{fl}(g) = |\hat{R}(g) - b| + b.^1$$

(2)

The gradient update of the flooded empirical risk with respect to the model parameters is performed as that of the empirical risk if $\hat{R}(g) > b$ and is otherwise performed in the opposite direction. The flooded empirical risk estimator is known to provide a better approximation of the risk than the empirical risk estimator in terms of MSE if the risk is greater than $b$.

For the mini-batched optimization, a gradient update of the flooded empirical risk is performed with respect to the mini-batch. Let $\hat{R}_t(g)$ denote the empirical risk with respect to the $t$-th mini-batch for $t \in \{1, 2, ..., T\}$. Then, by Jensen's inequality,

$$\hat{R}^{fl}(g) \leq \frac{1}{T}\sum_{t=1}^{T}(|\hat{R}_t(g) - b| + b).$$

(3)

Therefore, the mini-batched optimization minimizes the upper bound of the flooded empirical risk.

## 3 Method

In this section, we propose a novel regularization called WaveBound which is specially-designed for time series forecasting. We first deal with the drawbacks of applying original flooding to time series forecasting and then introduce a more desirable form of regularization.

### 3.1 Flooding in Time Series Forecasting

We first discuss how the original flooding may not effectively work for the time series forecasting problem. We start with rewriting Equation (2) using the risks at each prediction step and feature:

$$\hat{R}^{fl}(g) = \left|\hat{R}(g) - b\right| + b = \left|\left(\frac{1}{MK}\sum_{j,k}\hat{R}_{jk}(g)\right) - b\right| + b.$$

(4)

Flooded empirical risk constrains the lower bound of the average of the empirical risk for all prediction steps and features by a constant value of $b$. However, for the multivariate regression model, this regularization does not independently bound each $\hat{R}_{jk}(g)$. As a result, the regularization effect is concentrated on output variables where $\hat{R}_{jk}(g)$ greatly varies in training.

In this circumstance, the modified version of flooding can be explored by considering the individual training loss for each time step and feature. This can be done by subtracting the flood level $b$ for each time step and feature as follows:

$$\hat{R}^{const}(g) = \frac{1}{MK}\sum_{j,k}\left(|\hat{R}_{jk}(g) - b| + b\right).$$

(5)

---

[1]The constant $b$ outside of absolute value brackets does not affect the gradient update, but make sure $\tilde{R}^{fl}(g) = \hat{R}(g)$ if $\hat{R}(g) > b$. This property is especially useful in the analysis of estimation error.

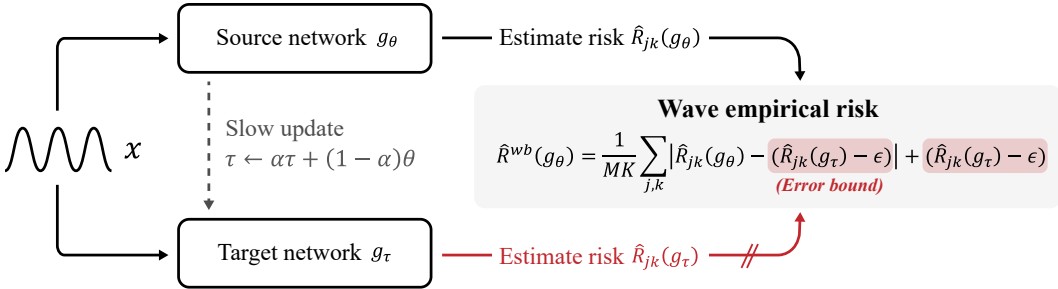

Figure 2: Our proposed WaveBound method provides the dynamic error bounds of the training loss for each time step and feature using the target network. The target network $g_\tau$ is updated with the EMA of the source network $g_\theta$. At $j$-th time step and $k$-th feature, the training loss is bounded by our estimated error bound $\hat{R}_{jk}(g_\tau) - \epsilon$, *i.e.*, the gradient ascent is performed instead of the gradient descent when the training loss is below the error bound.

For the remainder of this study, we denote this version of flooding as *constant flooding*. Compared to the original flooding that considers the average of the whole training loss, constant flooding individually constrains the lower bound of the training loss at each time step and feature by the value of $b$.

Nonetheless, it still fails to consider different difficulties of predictions for different mini-batches. For constant flooding, it is challenging to properly minimize the variants of empirical risk in the batch-wise training process. As in Equation 3, the mini-batched optimization minimizes the upper bound of the flooded empirical risk. The problem is that the inequality becomes less tight as each flooded risk term $\hat{R}_t(g) - b$ for $t \in \{1, 2, ..., T\}$ differs significantly. Since the time series data typically contains lots of unpredictable noise, this happens frequently as the loss of each batch highly varies. To ensure the tightness of the inequality, the bound for $\hat{R}_t(g)$ should be adaptively chosen for each batch.

## 3.2 WaveBound

As previously mentioned, to properly bound the empirical risk in time series forecasting, the regularization method should be considered with the following conditions: (i) The regularization should consider the empirical risk for each time step and feature individually. (ii) For different patterns, *i.e.*, mini-batches, different error bounds should be searched in the batch-wise training process. To handle this, we find the error bound for each time step and feature and dynamically adjust it at each iteration. Since manually searching different bounds for each time step and feature at every iteration is impractical, we estimate the error bounds for different predictions using the exponential moving average (EMA) model [13].

Concretely, two networks are employed throughout the training phase: the source network $g_\theta$ and target network $g_\tau$ which have the same architecture, but different weights $\theta$ and $\tau$, respectively. The target network estimates the proper lower bounds of errors for the predictions of the source network, and its weights are updated with the exponential moving average of the weights of the source network:

$$\tau \leftarrow \alpha\tau + (1 - \alpha)\theta, \tag{6}$$

where $\alpha \in [0, 1]$ is a target decay rate. On the other hands, the source network updates their weights $\theta$ using the gradient descent update in the direction of the gradient of *wave empirical risk* $\hat{R}^{wb}(g_\theta)$ which is defined as

$$\hat{R}^{wb}(g_\theta) = \frac{1}{MK} \sum_{j,k} \hat{R}^{wb}_{jk}(g_\theta),$$
$$\hat{R}^{wb}_{jk}(g_\theta) = \left| \hat{R}_{jk}(g_\theta) - (\hat{R}_{jk}(g_\tau) - \epsilon) \right| + (\hat{R}_{jk}(g_\tau) - \epsilon), \tag{7}$$

where $\epsilon$ is a hyperparameter indicating how far the error bound of the source network can be from the error of the target network. Intuitively, the target network guides the lower bound of the training loss for each time step and feature to prevent the source network from training towards a loss lower

than that bound, *i.e.*, overfitting to a certain pattern. As the exponential moving average model is known to have the effect of ensembling the source networks and memorizing training data visible in earlier iterations [13], the target network can robustly estimate the error bound of the source network against the instability mostly caused by noisy input data. Figure 2 shows how the source network and the target network perform in our WaveBound method. A summary of WaveBound is provided in Appendix B.

**Mini-batched optimization.** For $t \in \{1, 2, ..., T\}$, let $(\hat{R}_t^{wb})_{jk}(g)$ and $(\hat{R}_t)_{jk}(g)$ denote the wave empirical risk and the empirical risk at $j$-th step and $k$-th feature relative to the $t$-th mini-batch, respectively. Given the target network $g^*$, by Jensen's inequality,

$$\hat{R}_{jk}^{wb}(g) \leq \frac{1}{T} \sum_{t=1}^{T} \left( \left| (\hat{R}_t)_{jk}(g) - (\hat{R}_t)_{jk}(g^*) + \epsilon \right| + (\hat{R}_t)_{jk}(g^*) - \epsilon \right) = \frac{1}{T} \sum_{t=1}^{T} (\hat{R}_t^{wb})_{jk}(g). \quad (8)$$

Therefore, the mini-batched optimization minimizes the upper bound of wave empirical risk. Note that if $g$ is close to $g^*$, the values of $(\hat{R}_t)_{jk}(g) - (\hat{R}_t)_{jk}(g^*) + \epsilon$ are similar across mini-batches, which gives a tight bound in Jensen's inequality. We expect the EMA update to work so that this condition is met, giving a tight upper bound for the wave empirical risk in the mini-batched optimization.

**MSE reduction.** We show that the MSE of our suggested wave empirical risk estimator can be smaller than that of the empirical risk estimator given an appropriate $\epsilon$.

**Theorem 1.** *Fix measurable functions $g$ and $g^*$. Let $I := \{(i,j) : i = 1, 2, ..., M, j = 1, 2, ..., K\}$, and let $J(\mathcal{X}) := \{(i,j) \in I : \hat{R}_{ij}(g) < \hat{R}_{ij}(g^*) - \epsilon\}$. If the following two conditions hold:*

*(a) $\forall (i,j), (k,l) \in I$ such that $(i,j) \neq (k,l)$, $\hat{R}_{ij}(g) - \hat{R}_{ij}(g^*) \perp \hat{R}_{kl}(g)$*

*(b) $\hat{R}_{ij}(g^*) < R_{ij}(g) + \epsilon$ for all $(i,j) \in J(\mathcal{X})$ almost surely,*

*then $MSE(\hat{R}(g)) \geq MSE(\hat{R}^{wb}(g))$. Given the condition (a), if we have $0 < \alpha$ such that $\alpha < R_{ij}(g) - \hat{R}_{ij}(g^*) + \epsilon$ for all $(i,j) \in J(\mathcal{X})$ almost surely, then*

$$MSE(\hat{R}(g)) - MSE(\hat{R}^{wb}(g)) \geq 4\alpha^2 \sum_{(i,j) \in I} \Pr[\alpha < \hat{R}_{ij}(g^*) - \hat{R}_{ij}(g) - \epsilon]. \quad (9)$$

*Proof.* Please see Appendix A. $\square$

Intuitively, Theorem 1 states that the MSE of the empirical risk estimator can be reduced when the following conditions hold: (i) The network $g^*$ has sufficient expressive power so that the loss difference between $g$ and $g^*$ at each output variable is unrelated to the loss at the other output variables in $g$. (ii) $\hat{R}_{ij}(g^*) - \epsilon$ likely lies in between $\hat{R}_{ij}(g)$ and $R_{ij}(g)$. It is preferable to have $g^*$ as the EMA model of $g$ since the training loss of the EMA model cannot be readily below the test loss of the model. Then, $\epsilon$ can be chosen as a fixed small value so that the training loss of the source model at each output variable can be closely bounded below by the test loss at that variable.

## 4 Experiments

### 4.1 WaveBound with Forecasting Models

In this section, we evaluate our WaveBound method on real-world benchmarks using various time series forecasting models, including the state-of-the-art models.

**Baselines.** Since our method can be easily applied to deep-learning-based forecasting models, we evaluate our regularization adopted by several baselines including the state-of-the-art method. For the multivariate setting, we select Autoformer [5], Pyraformer [6], Informer [7], LSTNet [14], and TCN [16]. For the univariate setting, we additionally include N-BEATS [15] for the baseline.

**Datasets.** We examine the performance of forecasting models in six real-world benchmarks. (1) The Electricity Transformer Temperature (ETT) [7] dataset contains two years of data from two separate counties in China with intervals of 1-hour level (ETTh1, ETTh2) and 15 minutes level (ETTm1, ETTm2) collected from electricity transformers. Each time step contains an oil temperature and

Table 1: The results of WaveBound in the multivariate setting. All results are averaged over 3 trials.

| Models | | Autoformer [5] | | | | Pyraformer [6] | | | | Informer [7] | | | | LSTNet [14] | | | |
|---|---|---|---|---|---|---|---|---|---|---|---|---|---|---|---|---|---|
| | | Origin | | w/ Ours | | Origin | | w/ Ours | | Origin | | w/ Ours | | Origin | | w/ Ours | |
| Metric | | MSE | MAE | MSE | MAE | MSE | MAE | MSE | MAE | MSE | MAE | MSE | MAE | MSE | MAE | MSE | MAE |
| ETTm2 | 96 | 0.262 | 0.326 | **0.204** | **0.285** | 0.363 | 0.451 | **0.281** | **0.386** | 0.376 | 0.477 | **0.334** | **0.429** | 0.455 | 0.511 | **0.268** | **0.368** |
| | 192 | 0.284 | 0.342 | **0.265** | **0.322** | 0.708 | 0.648 | **0.624** | **0.599** | 0.751 | 0.672 | **0.698** | **0.631** | 0.706 | 0.660 | **0.464** | **0.508** |
| | 336 | 0.338 | 0.374 | **0.320** | **0.356** | 1.130 | 0.846 | **1.072** | **0.829** | 1.440 | 0.917 | **1.087** | **0.845** | 1.161 | 0.868 | **0.781** | **0.695** |
| | 720 | 0.446 | 0.435 | **0.413** | **0.408** | 2.995 | 1.386 | **1.917** | **1.119** | 3.897 | 1.498 | **2.984** | **1.411** | 3.288 | 1.494 | **2.312** | **1.239** |
| ECL | 96 | 0.202 | 0.317 | **0.176** | **0.288** | 0.256 | 0.360 | **0.241** | **0.345** | 0.335 | 0.417 | **0.289** | **0.378** | 0.268 | 0.366 | **0.185** | **0.291** |
| | 192 | 0.235 | 0.340 | **0.205** | **0.317** | 0.272 | 0.378 | **0.256** | **0.360** | 0.341 | 0.426 | **0.298** | **0.388** | 0.277 | 0.375 | **0.197** | **0.304** |
| | 336 | 0.247 | 0.351 | **0.217** | **0.327** | 0.278 | 0.383 | **0.269** | **0.371** | 0.369 | 0.448 | **0.305** | **0.394** | 0.284 | 0.382 | **0.217** | **0.326** |
| | 720 | 0.270 | 0.371 | **0.260** | **0.359** | 0.291 | 0.385 | **0.283** | **0.377** | 0.396 | 0.457 | **0.311** | **0.398** | 0.316 | 0.404 | **0.250** | **0.350** |
| Exchange | 96 | 0.153 | 0.285 | **0.146** | **0.274** | **0.604** | **0.624** | 0.615 | 0.627 | 0.979 | 0.791 | **0.878** | **0.765** | 0.483 | 0.518 | **0.357** | **0.432** |
| | 192 | 0.297 | 0.397 | **0.262** | **0.373** | 0.982 | 0.806 | **0.953** | **0.797** | 1.147 | **0.854** | **1.136** | 0.859 | 0.706 | 0.646 | **0.621** | **0.593** |
| | 336 | 0.438 | 0.490 | **0.425** | **0.483** | 1.264 | **0.934** | **1.263** | 0.944 | 1.592 | 1.014 | **1.461** | **0.992** | 1.055 | 0.800 | **0.837** | **0.691** |
| | 720 | 1.207 | 0.860 | **1.088** | **0.810** | 1.663 | 1.051 | **1.562** | **1.016** | 2.540 | 1.306 | **2.496** | **1.294** | 2.198 | 1.127 | **1.374** | **0.894** |
| Traffic | 96 | 0.645 | 0.399 | **0.596** | **0.352** | 0.635 | 0.364 | **0.622** | **0.341** | 0.731 | 0.412 | **0.671** | **0.364** | 0.735 | 0.446 | **0.587** | **0.356** |
| | 192 | 0.644 | 0.407 | **0.607** | **0.370** | 0.658 | 0.376 | **0.646** | **0.355** | 0.751 | 0.422 | **0.666** | **0.360** | 0.750 | 0.446 | **0.595** | **0.365** |
| | 336 | 0.625 | 0.390 | **0.603** | **0.361** | 0.668 | 0.377 | **0.653** | **0.355** | 0.822 | 0.465 | **0.709** | **0.387** | 0.778 | 0.455 | **0.623** | **0.378** |
| | 720 | 0.650 | 0.398 | **0.642** | **0.383** | 0.698 | 0.390 | **0.672** | **0.364** | 0.957 | 0.539 | **0.764** | **0.421** | 0.815 | 0.470 | **0.648** | **0.383** |
| Weather | 96 | 0.294 | 0.355 | **0.227** | **0.296** | 0.235 | 0.321 | **0.193** | **0.272** | 0.378 | 0.428 | **0.355** | **0.415** | 0.237 | 0.310 | **0.202** | **0.275** |
| | 192 | 0.308 | 0.368 | **0.283** | **0.340** | 0.340 | 0.415 | **0.306** | **0.372** | 0.462 | 0.467 | **0.424** | **0.448** | 0.277 | 0.343 | **0.254** | **0.316** |
| | 336 | 0.364 | 0.396 | **0.335** | **0.370** | 0.453 | 0.484 | **0.403** | **0.441** | 0.575 | 0.535 | **0.506** | **0.484** | 0.326 | 0.378 | **0.309** | **0.358** |
| | 720 | 0.426 | 0.433 | **0.401** | **0.411** | 0.599 | 0.563 | **0.535** | **0.519** | 1.024 | 0.751 | **0.972** | **0.712** | 0.412 | 0.431 | **0.398** | **0.415** |
| ILI | 24 | 3.468 | 1.299 | **3.118** | **1.200** | 4.822 | 1.489 | **4.679** | **1.459** | 5.356 | 1.590 | **4.947** | **1.494** | 7.934 | 2.091 | **6.331** | **1.816** |
| | 36 | 3.441 | 1.273 | **3.310** | **1.240** | 4.831 | **1.479** | **4.763** | 1.483 | 5.131 | 1.569 | **5.027** | **1.537** | 8.793 | 2.214 | **6.560** | **1.848** |
| | 48 | 3.086 | 1.184 | **2.927** | **1.128** | 4.789 | 1.465 | **4.524** | **1.439** | 5.150 | 1.564 | **4.920** | **1.514** | 7.968 | 2.068 | **6.154** | **1.779** |
| | 60 | 2.843 | 1.136 | **2.785** | **1.116** | 4.876 | 1.495 | **4.573** | **1.465** | 5.407 | 1.604 | **5.013** | **1.528** | 7.387 | 1.984 | **6.119** | **1.758** |

Table 2: The results of WaveBound in the univariate setting. All results are averaged over 3 trials.

| Models | | Autoformer [5] | | | | Pyraformer [6] | | | | Informer [7] | | | | N-BEATS [15] | | | |
|---|---|---|---|---|---|---|---|---|---|---|---|---|---|---|---|---|---|
| | | Origin | | w/ Ours | | Origin | | w/ Ours | | Origin | | w/ Ours | | Origin | | w/ Ours | |
| Metric | | MSE | MAE | MSE | MAE | MSE | MAE | MSE | MAE | MSE | MAE | MSE | MAE | MSE | MAE | MSE | MAE |
| ETTm2 | 96 | 0.098 | 0.239 | **0.085** | **0.221** | 0.078 | 0.209 | **0.070** | **0.197** | 0.085 | 0.224 | **0.081** | **0.218** | 0.073 | 0.198 | **0.067** | **0.188** |
| | 192 | 0.130 | 0.277 | **0.116** | **0.262** | 0.114 | 0.257 | **0.110** | **0.256** | 0.122 | 0.273 | **0.118** | **0.270** | 0.107 | 0.246 | **0.103** | **0.241** |
| | 336 | 0.162 | 0.311 | **0.143** | **0.293** | 0.178 | 0.325 | **0.153** | **0.306** | 0.153 | **0.304** | **0.148** | 0.305 | 0.163 | 0.310 | **0.135** | **0.284** |
| | 720 | 0.194 | 0.344 | **0.188** | **0.338** | 0.198 | 0.351 | **0.169** | **0.329** | 0.196 | 0.351 | **0.189** | **0.349** | 0.263 | 0.402 | **0.188** | **0.340** |
| ECL | 96 | 0.462 | 0.502 | **0.447** | **0.496** | 0.240 | 0.351 | **0.229** | **0.347** | 0.266 | 0.371 | **0.261** | **0.369** | 0.304 | 0.382 | **0.298** | **0.378** |
| | 192 | 0.557 | 0.565 | **0.515** | **0.538** | 0.262 | 0.367 | **0.253** | **0.365** | 0.283 | 0.385 | **0.281** | **0.383** | 0.323 | 0.396 | **0.322** | **0.395** |
| | 336 | 0.613 | 0.593 | **0.531** | **0.543** | 0.285 | 0.386 | **0.283** | **0.386** | 0.338 | 0.428 | **0.332** | **0.426** | 0.385 | 0.430 | **0.369** | **0.422** |
| | 720 | 0.691 | 0.632 | **0.604** | **0.591** | 0.309 | **0.411** | **0.307** | 0.415 | 0.631 | 0.612 | **0.378** | **0.463** | 0.462 | 0.487 | **0.433** | **0.473** |

6 additional features. (2) The Electricity (ECL) [2] dataset comprises 2 years of hourly electricity consumption of 321 clients. (3) The Exchange [14] dataset provides a collection of eight distinct countries on a daily basis. (4) The Traffic [3] dataset contains hourly statistics of various sensors in San Francisco Bay provided by the California Department of Transportation. Road occupancy rate is expressed as a real number between 0 and 1. (5) The Weather [4] dataset records 4 years of data (2010-2013) of 21 meteorological indicators collected at around 1,600 landmarks in the United States. (6) The ILI [5] dataset contains data from the Centers for Disease Control and Prevention's weekly reported influenza-like illness patients from 2002 to 2021, which describes the ratio of patients seen with ILI to the overall number of patients. As in Autoformer [5], we set $L = 36$ and $M \in \{24, 36, 48, 60\}$ for the ILI dataset, and set $L = 96$ and $M \in \{96, 192, 336, 720\}$ for the other datasets. We split each dataset into train/validation/test as follows: 6:2:2 ratio for the ETT dataset and 7:1:2 ratio for the rest.

**Multivariate Results.** Table 1 shows the performance of our method in terms of mean squared error (MSE) and mean absolute error (MAE) in the multivariate setting. We can observe that our

---

[2]https://archive.ics.uci.edu/ml/datasets/ElectricityLoadDiagrams20112014

[3]http://pems.dot.ca.gov

[4]https://www.ncei.noaa.gov/data/local-climatological-data/

[5]https://gis.cdc.gov/grasp/fluview/fluportaldashboard.html

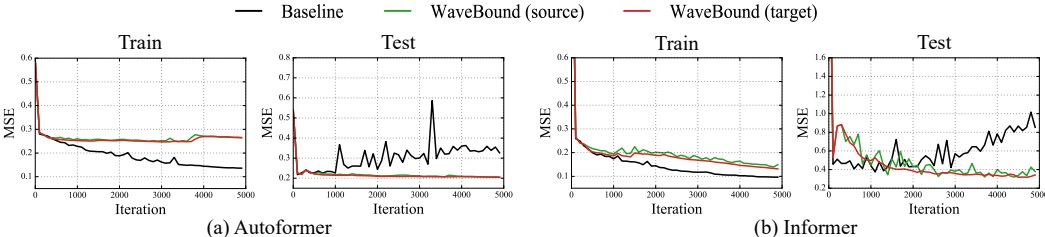

Figure 3: The training curves of models with and without WaveBound on the ETTm2 dataset. Without WaveBound, the training loss of both models decreases, but the test loss increases (See black lines), which indicates that both models tend to overfit at the training data. In contrast, the test loss of models with WaveBound continue to decrease after learning for even more epochs.

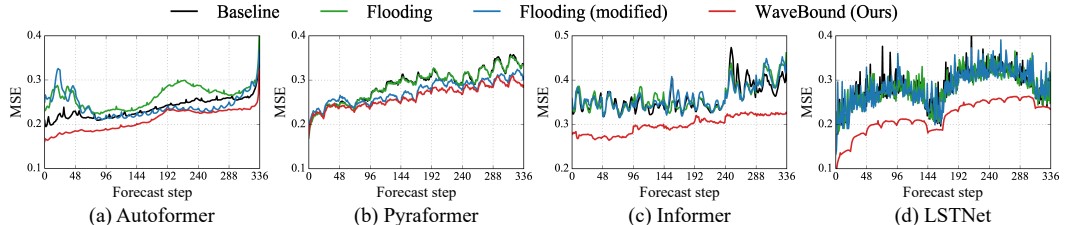

Figure 4: The test error of models trained with different regularization methods on the ECL dataset. Compared with original flooding and constant flooding, the test error of WaveBound is consistently lower at all time steps, which indicates that our method successfully improves generalization regardless of the range of predictions.

method consistently shows improvements for various forecasting models including the state-of-the-art methods. Notably, WaveBound improves both MAE and MSE of Autoformer on the ETTm2 dataset by **22.13%** ($0.262 \rightarrow 0.204$) in MSE and **12.57%** ($0.326 \rightarrow 0.285$) in MAE when $M = 96$. In particular, the performance is improved by **41.10%** ($0.455 \rightarrow 0.268$) in MSE and **27.98%** ($0.511 \rightarrow 0.368$) in MAE for LSTNet. For long-term ETTm2 forecasting settings ($M = 720$), WaveBound improves the performance of Autoformer by **7.39%** ($0.446 \rightarrow 0.413$) in MSE and **6.20%** ($0.435 \rightarrow 0.408$) in MAE. In all experiments, our method consistently shows performance improvements with various forecasting models. The results of full baselines and benchmarks are reported in Appendix D.

**Univariate Results.** WaveBound also shows superior results in the univariate setting, as reported in Table 2. In particular, for N-BEATS, which is designed especially for univariate time series forecasting, our method improves the performance on the ETTm2 dataset by **8.22%** ($0.073 \rightarrow 0.067$) in MSE and **5.05%** ($0.198 \rightarrow 0.188$) in MAE, when $M = 96$. For the ECL dataset, Informer with WaveBound shows improvements of **40.10%** ($0.631 \rightarrow 0.378$) in MSE and **24.35%** ($0.612 \rightarrow 0.463$) in MAE when $M = 720$. The results of the full baselines and benchmarks are reported in Appendix D.

**Generalization Gaps.** To identify overfitting, the generalization gap, which is the difference between the training loss and the test loss, can be examined. To verify that our regularization truly prevents overfitting, we depict both the training loss and test loss for models with and without WaveBound in Figure 3. Without WaveBound, the test loss starts to increase abruptly, showing a high generalization gap. In contrast, when using WaveBound, we can observe that the test loss continued to decrease, which indicates that WaveBound successfully addresses overfitting in time series forecasting.

## 4.2 Significance of Dynamically Adjusting Error Bounds

In WaveBound, the error bound is dynamically adjusted at each iteration for each time step and feature. To validate the significance of such dynamics, we compare our WaveBound with the original flooding and constant flooding which use the constant values for flood levels, as introduced in Section 3.

Table 3 compares the performance of variants of flooding regularization with different surrogates to empirical risk. The original flooding bounds the empirical risk by a constant while constant flooding bounds the risk at each feature and time step independently. We searched the flood level $b$ for the

Table 3: The results of variants of flooding regularization on the ECL dataset. We compare the forecasting accuracy when training the source network using different surrogates to empirical risk. All results are averaged over 3 trials and the constant value $b$ is faithfully searched in $\{0.00, 0.02, 0.04, ...0.40\}$.

| Method | Estimated risk (w/o constant) | | Autoformer | | Pyraformer | | Informer | | LSTNet | |
|---|---|---|---|---|---|---|---|---|---|---|
| | | | 96 | 336 | 96 | 336 | 96 | 336 | 96 | 336 |
| Base model | $\hat{R}(g)$ | MSE | 0.202 | 0.247 | 0.256 | 0.278 | 0.335 | 0.369 | 0.268 | 0.284 |
| | | MAE | 0.317 | 0.351 | 0.360 | 0.383 | 0.417 | 0.448 | 0.366 | 0.382 |
| Flooding [10] | $|\hat{R}(g) - b|$ | MSE | 0.194 | 0.247 | 0.257 | 0.277 | 0.335 | 0.368 | 0.268 | 0.284 |
| | | MAE | 0.309 | 0.351 | 0.360 | 0.382 | 0.416 | 0.447 | 0.366 | 0.381 |
| Constant flooding | $\frac{1}{MK} \sum_{j,k} |\hat{R}_{jk}(g) - b|$ | MSE | 0.198 | 0.247 | 0.257 | 0.277 | 0.333 | 0.369 | 0.268 | 0.284 |
| | | MAE | 0.314 | 0.351 | 0.360 | 0.382 | 0.415 | 0.448 | 0.366 | 0.382 |
| WaveBound (Avg.) | $|\hat{R}(g) - \hat{R}(g^*) + \epsilon|$ | MSE | 0.194 | 0.221 | 0.248 | 0.288 | 0.302 | 0.322 | 0.208 | 0.246 |
| | | MAE | 0.309 | 0.331 | 0.352 | 0.388 | 0.388 | 0.407 | 0.314 | 0.356 |
| **WaveBound (Indiv.)** | $\frac{1}{MK} \sum_{j,k} |\hat{R}_{jk}(g) - \hat{R}_{jk}(g^*) + \epsilon|$ | MSE | **0.176** | **0.217** | **0.241** | **0.269** | **0.289** | **0.305** | **0.185** | **0.217** |
| | | MAE | **0.288** | **0.327** | **0.345** | **0.371** | **0.378** | **0.394** | **0.291** | **0.326** |

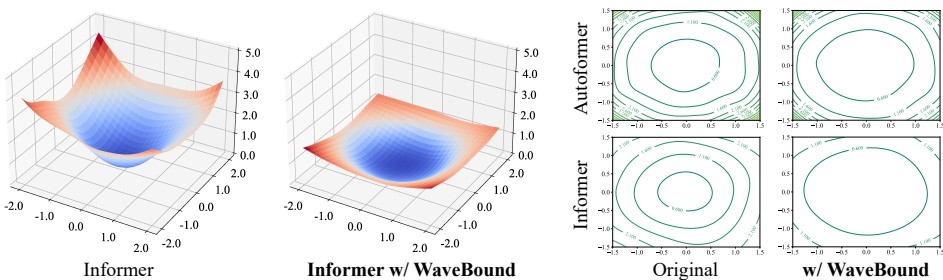

Figure 5: The loss landscapes of Autoformer and Informer trained with and without our WaveBound on the ETTh1 dataset. WaveBound flattens the loss landscapes for both models, improving the generalization of models.

regularization methods with constant value $b$ in space of $\{0.00, 0.02, 0.04, ...0.40\}$. As we expected, we cannot achieve the improvements when using a fixed constant value. The models trained by individually bounding the error in each output variable outperform other baselines by a large margin, which concretely shows the effectiveness of our proposed WaveBound method. The test error of different methods for each time step is depicted in Figure 4. For all time steps, WaveBound shows an improved generalization compared to original flooding and constant flooding, which highlights the significance of adjusting the error bounds in time series forecasting.

### 4.3   Flatness of Loss Landscapes

The visualization of loss landscapes [17] is introduced to evaluate how the model adequately generalizes. It is known that the flatter the loss landscapes of the model, the better the robustness and generalization [18, 19]. In this section, we depict the loss landscapes of models trained with and without WaveBound. Figure 5 shows the loss landscapes of Autoformer and Informer. We visualize the loss landscapes using filter normalization [17] and evaluate the MSE for every model for a fair comparison. We can observe that the model with WaveBound shows smoother loss landscapes compared to that of the original model. In other words, WaveBound flattens the loss landscapes of time series forecasting models and stabilizes the training.

## 5   Related Work

**Time series forecasting.** For time series forecasting tasks, various approaches have been proposed based on different principles. Statistical approaches can provide interpretability as well as a theoretical guarantee. Auto-regressive Integrated Moving Average  [20] and Prophet [21] are the most representative methods for statistical approaches. Another important class of time series forecasting is the state space models [22, 23] (SSMs). SSMs incorporate structural assumptions into the model

and learn latent dynamics of the time series data. However, due to its superior results in long-range forecasting, deep-learning-based approaches are mainly considered as the prominent solution for time series forecasting. To model the temporal dependencies in time series data, recurrent neural networks (RNN) [24, 25, 26, 1] and convolutional neural networks (CNN) [14, 27] are introduced in time series forecasting. Temporal convolutional networks (TCN) [28, 16, 29] are also considered for modeling temporal causality. Approaches combining SSMs and neural networks have also been proposed. DeepSSM [30] estimates state space parameters using RNN. Linear latent dynamics have been efficiently modeled using a Kalman filter [31, 32], and methodologies to model non-linear state variables have been proposed [33]. Other recent approaches include using SSMs with Rao-blackwellised particle filters [34] or SSMs with a duration switching mechanism [35].

Recently, transformer-based models have been introduced in time series forecasting due to their ability to capture the long-range dependencies. However, applying a self-attention mechanism increases the complexity of sequence length $L$ from $O(L)$ to $O(L^2)$. To alleviate the computational burden, several attempts such as LogTrans [8], Reformer [9], and Informer [7] re-designed the self-attention mechanism to a sparse version and reduced the complexity of the transformer. Haixu *et al.* [5] proposed the decomposition architecture with an auto-correlation mechanism called Autoformer to provide the series-wise connections. To model the temporal dependencies of different ranges, the pyramidal attention module is proposed in Pyraformer [6]. However, we observe that these models still fail to generalize due to the training strategy that enforces models to fit to all inconsistent patterns appearing in real data. In this work, we mainly focus on providing the adequate error bounds to prevent models from being overfitted to a certain pattern in the training procedure.

**Regularization methods.** Overfitting is one of the critical concerns for the over-parameterized deep networks. This can be identified by the generalization gap, which is the gap between the training loss and the test loss. To prevent overfitting and improve generalization, several regularization methods have been proposed. Decaying weight parameters [36], early stopping [37], and Dropout [38] have been commonly applied to avoid the high bias of deep networks. In addition to these methods, regularization methods specially designed for time series forecasting have also been proposed [39, 40].

Recently, the flooding [10] has been introduced to explicitly prevent *zero training loss*. By providing the lower bound of training loss, called the *flood level*, flooding allows the model not to completely fit to the training data, thus improving the generalization capacity of the model. In this work, we also attempt to tackle the zero training loss in time series forecasting. However, we find that bounding the average loss in time series forecasting does not perform as well as expected. In time series forecasting, an appropriate error bound for each feature and time step should be carefully chosen. In addition, a constant flood level may not be suitable to time series forecasting where the difficulty of prediction changes for every iteration in the mini-batch training process. To handle such issues, we suggest a novel regularization which fully considers the nature of time series forecasting.

# 6 Conclusion

In this work, we propose a simple yet effective regularization method called *WaveBound* for time series forecasting. WaveBound provides dynamic error bounds for each time step and feature using the slow-moving average model. With the extensive experiments on real-world benchmarks, we show that our regularization scheme consistently improves the existing models including the state-of-the-art model, addressing overfitting in time series forecasting. We also examine the generalization gaps and loss landscapes to discuss the effect of WaveBound in the training of over-parameterized networks. We believe that the significant performance improvements of our method indicate that regularization should be designed specifically for time series forecasting. We believe that our work will provide a meaningful insight into future research.

## Acknowledgments and Disclosure of Funding

This work was supported by the Institute of Information & communications Technology Planning & Evaluation (IITP) grant funded by the Korean government (MSIT) (No. 2019-0-00075, Artificial Intelligence Graduate School Program (KAIST)) and the National Research Foundation of Korea (NRF) grant funded by the Korean government (MSIT) (No. NRF-2022R1A2B5B02001913 and 2021H1D3A2A02096445). This work was also partially supported by NAVER Corp.

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
