# OpenReview forum: "WaveBound: Dynamic Error Bounds for Stable Time Series Forecasting"
_NeurIPS.cc/2022/Conference — NeurIPS 2022 Accept_

### Official Review · Reviewer_TFyo · 2022-07-06

**Rating:** 7
**Confidence:** 4
**Soundness:** 3 good
**Presentation:** 3 good
**Contribution:** 3 good

**Summary:**

This paper presents a regularization method for time series modeling aimed at preventing overfitting in deep learning methods. This method extends flooding regularization, which uses a constant target training risk, to use an exponential moving average of past models, yielding a level of training risk that is adaptive. The authors demonstrate that their improved formulation leads to better performance when using SOTA models (Autoformer, Pyraformer) on popular benchmarks datasets in the field (ett*, electricity, exchange).

**Questions:**

I’ve already enumerated some questions in the above section, but I include a few more questions/comments below:

It’s not clear to me why the design of the approach is actually tailored for time series data. You note that (1) vanilla flooding doesn’t deal well with inputs that have temporal and channel dimensions, and (2) vanilla flooding doesn’t gracefully handle inputs with lower signal-to-noise. (1) seems general to any sequence-to-sequence task, and I’m not sure if the method actually addresses (2) directly. If as practitioners we are worried about irreducible noise, why is the proposed adaptive method favorable over a fixed level of risk? The slow EMA network is also capable of overfitting to random noise.


**Limitations:**

The paper suggests that stochasticity is a source of overfitting in applications of deep learning to time series, but the proposed method doesn’t really deal with stochasticity explicitly, which I think is potentially a major shortcoming. In many of the benchmark datasets used for evaluation, the signal-to-noise ratio is quite small (e.g. exchange rates data) and any deterministic method is going to struggle to fit the data. In some sense the objective is simply misspecified. Beyond simply trying to limit overfitting to this noise with regularization, explicitly modeling the stochasticity can probably be very helpful.

**Strengths And Weaknesses:**

Strengths:
- The presentation is generally pretty clear and I found the paper easy to read
- The evaluation shows that the method yields consistent improvements over the chosen baselines
- The proposed method seems to be quite general purpose, and could probably be applied to other domains. The method essentially seems like an extension of vanilla flooding to sequence-to-sequence models.

Weaknesses:
- Section 3.2 could be better motivated. It’s not immediately obvious why bounding the risk to be similar for g and g* is desirable a priori or why it might lead to similar outcomes as vanilla flooding. Theorem 1, especially, should be motivated more, as it’s not clear what it implies practically. Do we just need to specify an appropriately sized epsilon and then suddenly we obtain lower MSEs across the board? Why wouldn’t we apply this to essentially every instance of ERM if there’s a MSE reduction? Clearly the conditions for the theorem to hold entail trade-offs that are not made obvious.
- It’s not entirely clear to me that this method works better than other obvious regularization methods. For example, in the plots shown in Figure 3 it seems like early stopping would be almost as effective as flooding for preventing overfitting and obtaining good test error. Moreover, it’s not clear to me that the proposed flooding method wouldn’t start overfitting as well if you continued to run it for more epochs. The EMA slow network prevents the loss from quickly going to zero, but eventually the EMA network can also begin overfitting.
- In general if the essential claim behind the paper is that overfitting is a major source of error in current deep learning methods for time series, it would be beneficial to investigate and support this claim more directly. For example you could examine how this phenomenon might change with dataset size or signal-to-noise ratio. Beyond proposing a specific technique, this analysis would be incredibly useful to the community, as deep learning still seems to struggle on time series and there’s no real consensus around what the most significant roadblocks are.

---

> ### Author Response · Authors · 2022-08-02
> **Response to Reviewer TFyo [3/3]**
>
> **[Q1] Benefits of using adaptive error bounds over the fixed level of bounds.**
>
> Reviewer TFyo highlights the important aspects of the motivation of WaveBound. As the reviewer mentioned, finding a proper fixed level of risk may be a prominent solution for preventing overfitting. Ideally, if one already knows how much irreducible noise is provided for each prediction during training, it may be optimal to give them accurate values of ‘noise’ to prevent the models from reducing the training loss below those levels.
>
> However, for at least two practical reasons, it is better to use adaptive error bounds than fixed levels of bounds. First, finding optimal fixed levels of risk for all different features and steps is inapplicable in practice as it introduces the high cost of hyperparameter searching. Another reason is that the fixed level of risk bounding may be inadequate in the mini-batched optimization. During training, the difficulty of prediction for each feature and time step is dynamically changed on a noisy dataset. We can easily assume the situation where the previous batch is given without any noise and perfectly reflects the seasonality of data, while the current batch contains the unpredictable noise caused by accidents. Ideally, the model should attempt to fit into the previous batch, but not into the current batch. However, if we use a fixed level of risk for bounding error, we bound error by equal value for both batches and then the model is likely to learn noise.
>
> To resolve such issues, we employ the EMA model to predict the error bound. The models updated with exponential moving average have an effect of ensembling various models from previous steps and known to be robust to the noise that unexpectedly appears in the dataset which are not observed in previous batches due to its memorization ability [1, 2]. Our WaveBound consistently proves its effectiveness with superior results compared to the other flooding variants. Despite the robustness of the EMA model, we agree that this network is also capable of overfitting to training data, but we think the overfitting problem can be alleviated by using our proposed WaveBound with other regularizations together. We will add the formal discussion to our main paper.
>
> ---
>
> **[Q2] Using deterministic methods as the main time series forecasting models.**
>
> As you mentioned, time series forecasting can be treated more appropriately by dealing with stochasticity. Several deep-learning-based time series forecasting models [3, 4, 5] attempt to directly deal with stochasticity in time series data. However, most state-of-the-art forecasting methods such as Autoformer and Informer for long-term time series forecasting tasks still rely on the deterministic method of learning. To validate the effectiveness of our WaveBound, we mainly used these SOTA baselines in our experiments and focused on the deterministic learning approaches.
>
> Nevertheless, we remark that WaveBound can be applied to a stochastic forecasting model which is capable of having overfitting problems. To verify this, we conduct additional experiments with DeepAR [3]. The table below shows the results of applying our WaveBound to DeepAR. In the experiments, we found that our WaveBound can be adapted to DeepAR, which is a stochastic forecasting model trained with the negative log-likelihood (NLL) loss and also improves the performance of the model. We believe that our WaveBound can be easily adapted to other stochasticity-based models and consistently improve their performance. The faithful consideration of our regularization method on the stochasticity-based forecasting method will be a prominent future direction.
>
> \begin{array}{lc| cc|cc}
> \hline
>  & & \rlap{\text{Origin}} &&\rlap{\text{w/ WaveBound}} \newline
> \hline
>  Metric & & NLL & MSE & NLL & MSE \newline
> \hline
> \ ECL   & 8   & 1.059\small{\pm0.152} & 0.447\small{\pm0.168} & \mathbf{0.935\small{\pm0.045}} & \mathbf{0.306\small{\pm0.001}} &  \newline
> \       & 16  & 1.168\small{\pm0.101} & 0.500\small{\pm0.208} & \mathbf{1.148\small{\pm0.052}} & \mathbf{0.371\small{\pm0.036}} &  \newline
> \       & 24  & 1.256\small{\pm0.073} & 0.617\small{\pm0.119} & \mathbf{1.180\small{\pm0.062}} & \mathbf{0.394\small{\pm0.043}} &  \newline
> \hline
> \hline
> \end{array}
>
> ---
> [1] Grill et al. “Bootstrap Your Own Latent - A New Approach to Self-Supervised Learning.”, NeurIPS, 2020.\
> [2] He et al. “Momentum Contrast for Unsupervised Visual Representation Learning.”, CVPR, 2020.\
> [3] Flunkert et al. “DeepAR: Probabilistic Forecasting with Autoregressive Recurrent Networks.”, CoRR abs/1704.04110, 2017.\
> [4] Rasul et al. “Multivariate Probabilistic Time Series Forecasting via Conditioned Normalizing Flows.”,  ICLR, 2021.\
> [5] Rasul et al. “Autoregressive Denoising Diffusion Models for Multivariate Probabilistic Time Series Forecasting.”, ICML, 2021

---

> > ### Comment · Reviewer_TFyo · 2022-08-07
> > **Author Response Rebuttal**
> >
> > Thank you for your very detailed response! I have raised my score from a weak accept to an accept.

---

> ### Author Response · Authors · 2022-08-02
> **Response to Reviewer TFyo [2/3]**
>
> **[C3] Justification of overfitting issues in time series forecasting.**
>
> Along with the experiments in the real-world benchmarks, we additionally conduct experiments with the synthetic dataset to simulate the overfitting issue in time series forecasting tasks. Concretely, we synthesize the univariate time series with the following formula:
> $2 \sin(2\pi   (t/32)) + \sin(2\pi   (t/48)) + \sigma  z$
> ,
> where $z$ is the normal distribution. From this example, we can validate how time series forecasting models can accurately predict the seasonal part of this series against unpredictable noise. As you suggested, we examine the performance of the forecasting model by varying the noise levels $\sigma$ and the size of the dataset $|\mathcal{D}|$. The table below shows the performance of Informer with and without WaveBound in this setting.
> As expected, the baseline showed lower performance as the given data became noisy and smaller, while our WaveBound improved the performance in all settings. This demonstrates that WaveBound can address overfitting issues in time series forecasting tasks. We also provide qualitative results in Appendix F.
>
> \begin{array}{l cc}
> \hline
>  & & \text{Informer} & \text{Informer w/ WaveBound} \newline
> \hline
> \ \sigma = 0.1 & MSE & 0.225\small{\pm0.024} & \mathbf{0.015\small{\pm0.001}}  \newline
> \              & MAE & 0.363\small{\pm0.017} & \mathbf{0.096\small{\pm0.003}}  \newline
> \hline
> \ \sigma = 0.3 & MSE & 0.316\small{\pm0.022} & \mathbf{0.102\small{\pm0.001}}  \newline
> \              & MAE & 0.441\small{\pm0.013} & \mathbf{0.252\small{\pm0.002}}  \newline
> \hline
> \ \sigma = 0.5 & MSE & 0.605\small{\pm0.061} & \mathbf{0.281\small{\pm0.001}}  \newline
> \              & MAE & 0.613\small{\pm0.029} & \mathbf{0.419\small{\pm0.000}}  \newline
> \hline
> \end{array}
>
>
> \begin{array}{l cc}
> \hline
>  & & \text{Informer} & \text{Informer w/ WaveBound} \newline
> \hline
> \ |\mathcal{D}| = 4000 & MSE & 0.311\small{\pm0.004} & \mathbf{0.246\small{\pm0.001}}  \newline
> \                      & MAE & 0.443\small{\pm0.003} & \mathbf{0.394\small{\pm0.000}}  \newline
> \hline
> \ |\mathcal{D}| = 3000 & MSE & 0.399\small{\pm0.003} & \mathbf{0.290\small{\pm0.002}}  \newline
> \                      & MAE & 0.505\small{\pm0.002} & \mathbf{0.435\small{\pm0.001}}  \newline
> \hline
> \ |\mathcal{D}| = 2000 & MSE & 0.605\small{\pm0.061} & \mathbf{0.281\small{\pm0.001}}  \newline
> \                      & MAE & 0.613\small{\pm0.029} & \mathbf{0.419\small{\pm0.000}}  \newline
> \hline
> \end{array}

---

> ### Author Response · Authors · 2022-08-02
> **Response to Reviewer TFyo [1/3]**
>
> **[C1] Clarifying the motivations of the theoretical explanation.**
>
> We deeply appreciate the constructive comments on this issue. We revise the explanation for mini-batched optimization and Theorem 1 in Section 3.2.
>
> **1. Why bounding the risk to be similar for $g$ and $g\*$ is desirable a priori?**
>
> Equation 8 in the paper shows that wave empirical risk at each output variable is bounded above by average wave empirical risk across mini-batches from Jensen’s inequality.
> If $g$ is close to $g^\ast$, the values of $(\hat{R}_t) _{jk}(g) - (\hat{R} _t) _{jk}(g^\ast) + \epsilon$ in the right side of Equation 8 becomes similar across mini-batches. Since Jensen’s inequality becomes tight as these values are similar, it gives a tight bound for the wave empirical risk in the mini-batched optimization. In the original flooding, the batch optimization may not properly minimize the flooded empirical risk since the empirical risk can vary significantly across mini-batches in the noisy dataset (Please refer to Equation 3).
>
> **2. Do we just need to specify an appropriately sized epsilon, and then suddenly we obtain lower MSEs across the board? Why wouldn’t we apply this to essentially every instance of ERM if there’s a MSE reduction?**
>
> We added an additional description of Theorem 1 in the MSE reduction section. The brief conclusion of Theorem 1 is that the wave empirical risk can be better than the empirical risk estimator in terms of MSE if $\hat{R} _{ij}(g^\ast) - \epsilon $ is likely to lie in between $\hat{R} _{ij}(g)$ and $R _{ij}(g)$. Under the assumption that the EMA model $g^\ast$ cannot be readily below the test loss of the model $g$, we can choose epsilon as a fixed small value (e.g., 0.01, 0.001) so that the training loss of $g$ at each output variable can be closely bounded below by the test loss at that variable. We may further reduce the MSE of the estimator if we use different $\epsilon$ for each output variable, but we did not attempt it due to the burden of hyperparameter searching.
>
> ---
>
> **[C2] How can proposed WaveBound cooperate with the existing regularization methods (e.g., early-stopping)?**
>
> First of all, our WaveBound aims to avoid reducing the training loss under the irreducible error for each output variable. This goal is complementary to that of other regularization techniques. In fact, since most time series forecasting baselines such as Autoformer adopt early-stopping in their training, we also use early-stopping in all experiments by default for fairness. Nevertheless, even when the early-stopping is applied for all experiments, we observe that our WaveBound consistently improves the performance of several baselines in all experiments. There are at least two possible reasons for this.
>
> First, early-stopping does not directly participate in the training of models. With the unstable models, it cannot help the model other than to finish training at an early stage. Unfortunately, due to the inconsistency of input, the instability of training is crucial for time series forecasting. In contrast, our WaveBound directly bounds the error and prevents models from fitting to the irreducible noise during training. In other words, it allows the training of the model to be steady and uninterrupted by early stopping. Note that our WaveBound also keeps the low test error even with the longer training iteration, as shown in Figure 3, which indicates that our proposed WaveBound can prevent the model from learning such noise even with long train iterations.
>
> The second reason mainly relies on the individual considerations for each feature and time step, which is the main contribution of our WaveBound. The criteria for early-stopping is the ‘mean’ of validation errors among all features and time steps. This means that overfitted features can largely affect the stop criterion of early stopping. Besides, when the scale of overfitted scales is relatively small, the overfitting of that feature cannot be prevented by using overfitting. On the contrary, our method allows the model to estimate the error bounds for features and time steps individually. That is, our WaveBound shows its specialty by taking into account that the prediction difficulty of each feature and time step is different in practice.
>
> To verify that consideration for each feature and time step is essential, we visualized the train curves for each feature and time step and confirmed that the test error increased for several features even before early-stopping was performed. We added this discussion in Appendix G.
>
> As you claimed, the EMA model can also begin overfitting as it runs for long epochs. Therefore, the most practical solution we think is to use WaveBound with other regularization techniques such as early-stopping together.

---

### Official Review · Reviewer_v7br · 2022-07-07

**Rating:** 7
**Confidence:** 4
**Soundness:** 3 good
**Presentation:** 3 good
**Contribution:** 4 excellent

**Summary:**

This work proposed WaveBound, a new regularization method to prevent unstable training and over-fitting for time series forecasting problems by encouraging a non-zero training loss. Compared with the original flooding bound, WaveBound consider the error bound for each individual feature and step. Besides, the error bound dynamically adjusts with the help of a target network. The paper show the efficacy of the method by extensive experiments.

**Questions:**

1. In table 3, it shows WaveBound outperforms both constant flooding and original flooding. However, it seems like constant flooding along is not improving. I’m curious how will the model perform if you delete the individual error feature from the WaveBound, say using a loss function similar to original flooding but b term is from a target network. Also in Fig 4, when you have a univariate problem and small prediction length, the wavebound still performs better than the baseline.

2. In Fig 3b, there’s a jump of testing MSE for WaveBound at the beginning of training? Do you have any explanation for that?

**Limitations:**

there's no potential negative societal impact

**Strengths And Weaknesses:**

Strongness:

1. The paper introduces a new way to regularize the error terms individually. The error bound is automatically customized.
2. The experiments show the improvement of this regularization methods over many SOTA models.

Weakness
1. I’m confused about the inspiration to allow individual flooding for each time step and feature. It is understandable that time series forecasting shall have different learning difficulty/error patterns for each output element. However, in the testing phase, using either MAE or MSE does not bias the error and metric is fair for all the elements. Is that possible the target network decreases the learning difficulty for certain locations, therefore for certain element, both target and source network produces bad prediction, therefore leading to overall larger testing error? By intuition, if one feature receives larger error (for instance less predictable), it’s natural to give more “bias” to that feature in optimization which is reflected from the loss function. For tasks like image classification, self-supervised way is fine because individual probability is bounded. In regression tasks using MAE or MSE error, the error magnitude could vary a lot among different features, so it seems counter-intuitive for me to massively balance the “bias” in training. I hope the author could elaborate this in the paper.

---

> ### Author Response · Authors · 2022-08-02
> **Response to Reviewer v7br [2/2]**
>
> **[Q1] How does the performance change if using dynamic bounds to the batch-wise error?**
>
> As you suggested, the main idea of our WaveBound, bounding the error of time series forecasting models with the target network, can be adopted for each batch, not for each feature and time step. To reflect your feedback, we examine the new variant of the WaveBound called WaveBound-B, which uses the target network for bounding the batch-wise MSE while not considering each feature and time step individually.
>
> The table below shows the performance of WaveBound-B along with our proposed version of WaveBound. As you expected, adaptively bounding the batch-wise error also improves the performance of the forecasting model. Nevertheless, our proposed WaveBound still shows the best performance among all the variants in the experiments. This indicates that our WaveBound brings advantages by considering the different difficulties of prediction of each feature and time step with the EMA model, which are consistent with our contributions. We will add these experiments and discussions to our Appendix.
>
> Furthermore, to verify whether the difficulty of predictions truly differs for each feature and time step, we conducted additional analysis in Appendix G. Specifically, by depicting the train curves for different features and time steps, we can observe that the trends of the train curves between each feature and time step are clearly different.
>
> \begin{array}{l c | cc | cc | cc }
> \hline
>         &     & \rlap{\text{Original}} && \rlap{\text{WaveBound-B}} && \rlap{\text{WaveBound}}& \newline
> \hline
> \ Metric  &     & MSE & MAE & MSE & MAE & MSE & MAE \newline
> \hline
> \ Autoformer & 96  & 0.202\small{\pm0.003} & 0.317\small{\pm0.003}& 0.194\small{\pm0.001} & 0.309\small{\pm0.001} &  \mathbf{0.176\small{\pm0.003}} & \mathbf{0.288\small{\pm0.003}}  \newline
> \            & 336 & 0.247\small{\pm0.011} & 0.351\small{\pm0.009}& 0.221\small{\pm0.009} & 0.331\small{\pm0.006} &  \mathbf{0.217\small{\pm0.006}} & \mathbf{0.327\small{\pm0.005}} \newline
> \hline
> \ Pyraformer & 96  & 0.256\small{\pm0.002} & 0.360\small{\pm0.001}& 0.248\small{\pm0.002} & 0.352\small{\pm0.002} &  \mathbf{0.241\small{\pm0.001}} & \mathbf{0.345\small{\pm0.001}} \newline
> \            & 336 & 0.278\small{\pm0.007} & 0.383\small{\pm0.006}& 0.288\small{\pm0.011} & 0.388\small{\pm0.010} &  \mathbf{0.269\small{\pm0.005}} & \mathbf{0.371\small{\pm0.005}} \newline
> \hline
> \ Informer   & 96  & 0.335\small{\pm0.008} & 0.417\small{\pm0.004}& 0.302\small{\pm0.004} & 0.388\small{\pm0.003} &  \mathbf{0.289\small{\pm0.003}} & \mathbf{0.378\small{\pm0.002}} \newline
> \            & 336 & 0.369\small{\pm0.011} & 0.448\small{\pm0.009}& 0.322\small{\pm0.010} & 0.407\small{\pm0.008} &  \mathbf{0.305\small{\pm0.008}} & \mathbf{0.394\small{\pm0.008}} \newline
> \hline
> \ LSTNet     & 96  & 0.268\small{\pm0.004} & 0.366\small{\pm0.003}& 0.208\small{\pm0.010} & 0.314\small{\pm0.010} &  \mathbf{0.185\small{\pm0.003}} & \mathbf{0.291\small{\pm0.003}} \newline
> \            & 336 & 0.284\small{\pm0.001} & 0.382\small{\pm0.002}& 0.246\small{\pm0.009} & 0.356\small{\pm0.009} &  \mathbf{0.217\small{\pm0.004}} & \mathbf{0.326\small{\pm0.005}} \newline
> \hline
> \hline
> \end{array}
>
> ---
>
> **[Q2] Jump of the test error in Figure 3.**
>
> Following the conventional implementation of EMA update, in the early learning stage (up to 300 iterations), EMA is not performed and the parameters of the source network are copied to that of the target network. We also do not bound the error of the source network in this stage. Therefore, we guess that the phenomenon is caused by the randomness of the source network learning. After the 300 iterations, the test error is continuously reduced by performing EMA updates. We appreciate you for bringing this to our attention.

---

> > ### Comment · Reviewer_v7br · 2022-08-09
> > **response to rebuttal**
> >
> > Thank you very much for the rebuttal response, my concerns are addressed. I will stand my position as accept.

---

> ### Author Response · Authors · 2022-08-02
> **Response to Reviewer v7br [1/2]**
>
> **[C1] Inspiration of individual bounding for each time step and feature.**
>
> First, flooding approaches aim to ensure that the model does not reduce its training loss under irreducible errors. In the time series forecasting, we expect that the bound should be searched differently for each output variable as they have different difficulties in prediction. As a consequence, we suggest using the EMA model to predict the appropriate error bounds for each output variable.
>
> If the target network sets the difficulty (error of the target network) for certain output variables too low, then training on those variables will be the same as the original MSE training. However, since the EMA model (target network) is well known to be robust against noisy data and has the effect of ensembling the source networks, we think that the situation concerned by the reviewer is unlikely to happen. With the appropriately selected epsilon, we expect that the training loss on each variable eventually stays close to the irreducible error on that variable. Therefore, this mechanism cannot readily interfere with the learning of certain variables and has the benefit of directly bounding training loss at each variable.

---

### Official Review · Reviewer_2uZk · 2022-07-10

**Rating:** 5
**Confidence:** 4
**Soundness:** 2 fair
**Presentation:** 3 good
**Contribution:** 2 fair

**Summary:**

This paper proposes an effective regularization method called WaveBound for time series forecasting, targeting on the overfitting problem of deep networks. WaveBound uses two networks throughout the training phase. The first network provides lower bounds of errors for the second network, which improves the generalization ability of the second network. Experimental results show that WaveBound improves the performance of several time series forecasting models.

**Questions:**

1. Is overfitting a real issue for the time-series forecasting task? Can WaveBound work properly on short-term forecasting tasks? Please consider evaluating WaveBound on short forecasting tasks, i.e, exchange rate, ECL, traffic used in the paper but forecast for 3, 6, 12 steps. Some spatial-temporal time-series forecasting tasks are also quite popular, e.g., PEMS08, 07, 03, 04. These experimental settings are well studied in the literature, and their corresponding SOTA models are more mature. If WaveBound works on these tasks, the soundness of this paper will be largely improved. [[1](https://dl.acm.org/doi/pdf/10.1145/3394486.3403118), [2](https://arxiv.org/pdf/2106.09305.pdf), [3](https://arxiv.org/pdf/2205.08689.pdf)]

2. What is the benefit of using the source network to learn the lower bound defined by the target network? EMA model is proven to have a positive impact on test-time performance in other areas, such as semi-supervised learning. What if we only EMA without flooding? Please conduct an ablation study.

3. How is WaveBound compared with RIN? Can we use these two methods together and achieve better results?

4. Simple models are also effective for the long-term forecasting task. Is WaveBound effective on small-scale models, e.g., MLP?

5. Since WaveBound runs two models at the same time, does WaveBound largely increase the computational and memory cost?

**Limitations:**

The computational efficiency of WaveBound is not discussed in the paper.

**Strengths And Weaknesses:**

Strength:

1. This work adapts a regularization technique called flooding to TSF tasks, and it improves the generalization ability of existing time-series forecasting models.

2. The proposed method seems to be simple yet effective.

Weakness:

1. The overfitting problem of existing TSF models is not well motivated. It is until the experimental section wherein Figures 3 and 4 show the training-testing error curves of some models used in the paper. It is essential to provide sound justifications that overfitting does exist in TFF models first.

2. In time-series forecasting, a large generalization loss can also come from the distribution shift between training data and test data.  The Reversible Instance Norm (RIN) technique proposed in ICLR'22 is a simple way to mitigate this issue. Since both RIN and WaveBound try to reduce generalization errors, please compare them. Also, can these two methods be used at the same time and how would they perform together?

3. The evaluations are performed on long-term forecasting tasks only. As the long-term forecasting problem itself is not well studied yet, it weakens the effectiveness of the proposed solution. Consequently, it is essential to evaluate WaveBound in some mature experimental settings, i.e, well-developed models for short-term forecasting tasks.

4. The proposed method heavily relies on the EMA technique, but no ablation study is performed from this aspect.

---

> ### Author Response · Authors · 2022-08-02
> **Response to Reviewer 2uZk [4/4]**
>
> **[Q4] Effectiveness of WaveBound on small-scale models (e.g., MLP).**
>
> As the reviewer claimed, simple architecture such as multi-layer perceptron (MLP) may be effective for long-term time series forecasting. In fact, we tested our WaveBound on a SOTA univariate forecasting model, N-Beats[3], which is composed of MLPs without complex modules such as self-attention layers. Table 2 in our paper shows the performance improvement when applying WaveBound to this model.
>
> Nevertheless, one may be curious about the performance of WaveBound on a simpler model than this baseline. For the interest of the reviewer, we additionally conduct experiments with the simple MLP model which consists of only three linear layers. The table below shows the performance of this model with and without WaveBound in the univariate setting. Our WaveBound consistently improves the performance of MLP-based models, and this highlights our method can be adopted by various architectures as the ‘model-agnostic’ regularization method. Additional experiments for the effect of WaveBound on MLP-based models (MLP) is updated in the Appendix K.
>
> \begin{array}{lc| cc|cc}
> \hline
>  & & \rlap{\text{Origin}} &&\rlap{\text{w/ WaveBound}} \newline
> \hline
>  Metric & & MSE & MAE & MSE & MAE \newline
> \hline
> \ ETTm2 & 96   & 0.071\small{\pm0.001} & 0.195\small{\pm0.002} & \mathbf{0.068\small{\pm0.000}} & \mathbf{0.191\small{\pm0.001}} &  \newline
> \       & 192  & 0.105\small{\pm0.004} & 0.243\small{\pm0.005} & \mathbf{0.104\small{\pm0.000}} & \mathbf{0.241\small{\pm0.001}} &  \newline
> \       & 336  & 0.141\small{\pm0.009} & 0.289\small{\pm0.010} & \mathbf{0.136\small{\pm0.001}} & \mathbf{0.284\small{\pm0.001}} &  \newline
> \       & 720  & 0.207\small{\pm0.010} & 0.357\small{\pm0.009} & \mathbf{0.194\small{\pm0.002}} & \mathbf{0.343\small{\pm0.002}} &  \newline
> \hline
> \ ECL   & 96   & 0.325\small{\pm0.003} & 0.401\small{\pm0.001} & \mathbf{0.317\small{\pm0.003}} & \mathbf{0.392\small{\pm0.002}} &  \newline
> \       & 192  & 0.334\small{\pm0.001} & 0.408\small{\pm0.001} & \mathbf{0.330\small{\pm0.004}} & \mathbf{0.401\small{\pm0.001}} &  \newline
> \       & 336  & 0.389\small{\pm0.007} & 0.446\small{\pm0.007} & \mathbf{0.380\small{\pm0.002}} & \mathbf{0.436\small{\pm0.001}} &  \newline
> \       & 720  & 0.441\small{\pm0.005} & 0.486\small{\pm0.002} & \mathbf{0.438\small{\pm0.002}} & \mathbf{0.481\small{\pm0.001}} &  \newline
> \hline
> \hline
> \end{array}
>
> ---
>
> **[Q5] Information on computational efficiency.**
>
> As you pointed out, WaveBound introduces additional computation and memory costs in the training phase as it employs the EMA model. To analyze the computation and memory cost of WaveBound, we train the baseline models with and without WaveBound on ETTm2 dataset and prediction length 96. We use a single TITAN RTX GPU for all experiments. The table below shows the computation time per epoch and maximum GPU memory occupied in the training phase. We also remark that WaveBound does not introduce additional computation and memory costs in the inference phase.
>
> \begin{array}{l|cc|cc}
> \hline
> \        & \text{Training Time (Sec/Epoch)} &&& \text{GPU Memory (GB)} &&&  \newline
> \hline
> \ Model   & \text{Origin} & \text{WaveBound} & \text{Origin} & \text{WaveBound} \newline
> \hline
> \ Autoformer & 67.722  & 84.075 & 2.13 & 2.20 \newline
> \ Pyraformer & 27.024  & 39.083 & 0.47 & 0.50 \newline
> \ Informer   & 75.251  & 97.154 & 1.80 & 1.86 \newline
> \ LSTNet     & 20.380  & 23.347 & 0.02 & 0.02 \newline
> \hline
> \end{array}
>
>
> ---
>
> [1] Kim et al. "Reversible instance normalization for accurate time-series forecasting against distribution shift.", ICLR, 2021.\
> [2] Liu et al. “Time Series is a Special Sequence: Forecasting with Sample Convolution and Interaction.”, CoRR abs/2106.09305, 2021.\
> [3] Oreshkin et al, “N-BEATS: Neural basis expansion analysis for interpretable time series forecasting.”, ICLR, 2020.

---

> ### Author Response · Authors · 2022-08-02
> **Response to Reviewer 2uZk [3/4]**
>
> **[C4, Q2] Ablation study of bounding errors in time series forecasting.**
>
> In WaveBound, the network updated with exponential moving average (EMA) is used to guide the lower bound of the predictions. In this circumstance, the reviewers pointed out that the discussion of the impact of EMA technique can be conducted for emphasizing the importance of error bounding. We agree with the observations and value your suggestions.
>
> Following the comments, we conduct an ablation study by using only the EMA model without flooding; using their results for predictions directly, not for bounding the errors of the source network. The table below shows the results of the ablation study. As expected by the reviewer, since EMA itself has a positive impact on the prediction, the EMA model without flooding shows performance improvements compared to the original baselines. However, WaveBound still outperforms this baseline with a considerable margin. We believe that these observations prove the effectiveness of dynamic error bounding. Ablation study on the EMA model is updated in Appendix J.
>
> \begin{array}{l c | cc | cc | cc }
> \hline
>         &     & \rlap{\text{Original}} && \rlap{\text{EMA w/o flooding}} && \rlap{\text{WaveBound (Ours)}}& \newline
> \hline
> \ Metric  &     & MSE & MAE & MSE & MAE & MSE & MAE \newline
> \hline
> \ Autoformer & 96  & 0.202\small{\pm0.003} & 0.317\small{\pm0.003} & 0.193\small{\pm0.001} & 0.308\small{\pm0.001} &  \mathbf{0.176\small{\pm0.003}} & \mathbf{0.288\small{\pm0.003}}  \newline
> \            & 336 & 0.247\small{\pm0.011} & 0.351\small{\pm0.009} & 0.224\small{\pm0.006} & 0.334\small{\pm0.003} &  \mathbf{0.217\small{\pm0.006}} & \mathbf{0.327\small{\pm0.005}} \newline
> \hline
> \ Pyraformer & 96  & 0.256\small{\pm0.002} & 0.360\small{\pm0.001} & 0.247\small{\pm0.002} & 0.351\small{\pm0.002} &  \mathbf{0.241\small{\pm0.001}} & \mathbf{0.345\small{\pm0.001}} \newline
> \            & 336 & 0.278\small{\pm0.007} & 0.383\small{\pm0.006} & 0.287\small{\pm0.009} & 0.388\small{\pm0.010} &  \mathbf{0.269\small{\pm0.005}} & \mathbf{0.371\small{\pm0.005}} \newline
> \hline
> \ Informer   & 96  & 0.335\small{\pm0.008} & 0.417\small{\pm0.004} & 0.305\small{\pm0.005} & 0.391\small{\pm0.006} &  \mathbf{0.289\small{\pm0.003}} & \mathbf{0.378\small{\pm0.002}} \newline
> \            & 336 & 0.369\small{\pm0.011} & 0.448\small{\pm0.009} & 0.326\small{\pm0.017} & 0.411\small{\pm0.013} &  \mathbf{0.305\small{\pm0.008}} & \mathbf{0.394\small{\pm0.008}} \newline
> \hline
> \ LSTNet     & 96  & 0.268\small{\pm0.004} & 0.366\small{\pm0.003} & 0.201\small{\pm0.002} & 0.311\small{\pm0.003} &  \mathbf{0.185\small{\pm0.003}} & \mathbf{0.291\small{\pm0.003}} \newline
> \            & 336 & 0.284\small{\pm0.001} & 0.382\small{\pm0.002} & 0.241\small{\pm0.004} & 0.352\small{\pm0.004} &  \mathbf{0.217\small{\pm0.004}} & \mathbf{0.326\small{\pm0.005}} \newline
> \hline
> \hline
> \end{array}

---

> ### Author Response · Authors · 2022-08-02
> **Response to Reviewer 2uZk [2/4]**
>
> **[C3, Q1] WaveBound for the short-term forecasting / spatial-temporal time series forecasting tasks.**
>
> As the reviewer suggested, we evaluate the performance of WaveBound on the short-term forecasting task. For this experiment, we use exchange rate, ECL, and traffic dataset and attempt to forecast for 3, 6, and 12 steps. The table below shows the performance of the solid baseline, Autoformer, with and without our WaveBound in this setting.
>
> Due to the short discussion period, we decided to focus on validating the short-term forecasting performance of the baseline in this period. To reflect the reviewer’s comments as much as possible, we are now examining our WaveBound for spatial-temporal forecasting models such as SCINet [2] and will add the results in the final version of the paper. The effect of WaveBound in short-term prediction tasks is updated in Appendix I. Thank you for your constructive comments.
>
> \begin{array}{lc| cc|cc}
> \hline
>  & & \rlap{\text{Origin}} &&\rlap{\text{w/ WaveBound}} \newline
> \hline
>  Metric & & MSE & MAE & MSE & MAE \newline
> \hline
> \ ECL & 3  & 0.148\small{\pm0.001} & 0.274\small{\pm0.001} & \mathbf{0.133\small{\pm0.002}} & \mathbf{0.256\small{\pm0.002}} &  \newline
> \     & 6  & 0.152\small{\pm0.001} & 0.277\small{\pm0.000} & \mathbf{0.140\small{\pm0.001}} & \mathbf{0.263\small{\pm0.001}} &  \newline
> \     & 12 & 0.157\small{\pm0.001} & 0.282\small{\pm0.002} & \mathbf{0.144\small{\pm0.001}} & \mathbf{0.265\small{\pm0.001}} &  \newline
> \hline
> \ Exchange & 3  & 0.034\small{\pm0.003} & 0.132\small{\pm0.005} & \mathbf{0.029\small{\pm0.009}} & \mathbf{0.121\small{\pm0.020}} &  \newline
> \          & 6  & \mathbf{0.030\small{\pm0.003}} & 0.126\small{\pm0.007} & \mathbf{0.030\small{\pm0.007}} & \mathbf{0.124\small{\pm0.016}} &  \newline
> \          & 12 & 0.039\small{\pm0.004} & 0.143\small{\pm0.007} & \mathbf{0.032\small{\pm0.005}} & \mathbf{0.129\small{\pm0.008}} &  \newline
> \hline
> \ Traffic & 3  & 0.555\small{\pm0.005} & 0.385\small{\pm0.007} & \mathbf{0.520\small{\pm0.002}} & \mathbf{0.352\small{\pm0.002}} &  \newline
> \         & 6  & 0.555\small{\pm0.001} & 0.377\small{\pm0.003} & \mathbf{0.531\small{\pm0.001}} & \mathbf{0.351\small{\pm0.003}} &  \newline
> \         & 12 & 0.557\small{\pm0.004} & 0.375\small{\pm0.004} & \mathbf{0.530\small{\pm0.006}} & \mathbf{0.343\small{\pm0.004}} &  \newline
> \hline
> \end{array}

---

> ### Author Response · Authors · 2022-08-02
> **Response to Reviewer 2uZk [1/4]**
>
> **[C1] Justification of the overfitting issues in time series forecasting.**
>
> Thank you for pointing out the matter. To provide an additional justification of the overfitting issues in the time series forecasting, we conducted additional experiments with the synthetic dataset in the section F of appendix. In the experiments, we demonstrate that the existing time series model tends to overfit with the noisy or small dataset, and WaveBound can alleviate that issue. We will also consider locating Figure 3 in the front of the paper to motivate the overfitting problem in the time series forecasting problem in the camera-ready version.
>
> ---
>
> **[C2, Q3] WaveBound with Reversible Instance Normalization (RevIN)**
>
> As you mentioned, the reversible instance normalization (RevIN) [1] is recently proposed to mitigate the discrepancy between the train data and the test data, while WaveBound aims to reduce generalization errors by providing the proper error bound for each output variable. Even though RevIN resolves the distribution shift problem, the model may overfit if it reduces the training loss below the irreducible error. This implies that WaveBound and RevIN can be used at the same time to enhance the generalization performance of the forecasting model. As shown in the table below, the performance of Autoformer and Informer is improved when both RevIN and WaveBound are applied. Along with RevIN, WaveBound improves the performance of models with a significant margin.
>
> \begin{array}{lc|cc cc cc|cc cc cc}
> \hline
>         &     & \text{Autoformer} &&&&&& \text{Informer} &&&&&&  \newline
> \hline
>         &     & \text{Original} && \text{RevIN} && \text{RevIN + WaveBound} && \text{Original} && \text{RevIN} && \text{RevIN+ WaveBound}& \newline
> \ Metric&     & MSE & MAE & MSE & MAE & MSE & MAE & MSE & MAE & MSE & MAE & MSE & MAE \newline
> \hline
> \ ETTm2 & 96  & 0.262\small{\pm0.037} & 0.326\small{\pm0.014} & 0.230\small{\pm0.007} & 0.303\small{\pm0.006} & \mathbf{0.208\small{\pm0.002}} & \mathbf{0.282\small{\pm0.002}} & 0.376\small{\pm0.056} & 0.477\small{\pm0.047} & 0.270\small{\pm0.041} & 0.326\small{\pm0.025} & \mathbf{0.206\small{\pm0.000}} & \mathbf{0.286\small{\pm0.001}}  \newline
> \       & 192 & 0.284\small{\pm0.003} & 0.342\small{\pm0.002} & 0.291\small{\pm0.005} & 0.338\small{\pm0.003} & \mathbf{0.268\small{\pm0.001}} & \mathbf{0.319\small{\pm0.001}} & 0.751\small{\pm0.020} & 0.672\small{\pm0.004} & 0.475\small{\pm0.078} & 0.439\small{\pm0.033} & \mathbf{0.293\small{\pm0.011}} & \mathbf{0.341\small{\pm0.008}}  \newline
> \       & 336 & 0.338\small{\pm0.007} & 0.374\small{\pm0.005} & 0.346\small{\pm0.004} & 0.369\small{\pm0.004} & \mathbf{0.329\small{\pm0.000}} & \mathbf{0.357\small{\pm0.000}} & 1.440\small{\pm0.180} & 0.917\small{\pm0.067} & 0.517\small{\pm0.089} & 0.467\small{\pm0.041} & \mathbf{0.367\small{\pm0.005}} & \mathbf{0.385\small{\pm0.003}}  \newline
> \       & 720 & 0.446\small{\pm0.011} & 0.435\small{\pm0.006} & 0.435\small{\pm0.010} & 0.419\small{\pm0.007} & \mathbf{0.417\small{\pm0.003}} & \mathbf{0.407\small{\pm0.001}} & 3.897\small{\pm0.562} & 1.498\small{\pm0.128} & 0.646\small{\pm0.075} & 0.531\small{\pm0.034} & \mathbf{0.476\small{\pm0.015}} & \mathbf{0.442\small{\pm0.007}}  \newline
> \hline
> \ ECL   & 96  & 0.202\small{\pm0.003} & 0.317\small{\pm0.003} & 0.178\small{\pm0.001} & 0.285\small{\pm0.001} & \mathbf{0.173\small{\pm0.008}} & \mathbf{0.276\small{\pm0.006}} & 0.335\small{\pm0.008} & 0.417\small{\pm0.004} & 0.196\small{\pm0.001} & 0.303\small{\pm0.000} & \mathbf{0.166\small{\pm0.001}} & \mathbf{0.269\small{\pm0.001}}  \newline
> \       & 192 & 0.235\small{\pm0.011} & 0.340\small{\pm0.009} & 0.218\small{\pm0.010} & 0.317\small{\pm0.009} & \mathbf{0.212\small{\pm0.017}} & \mathbf{0.308\small{\pm0.015}} & 0.341\small{\pm0.013} & 0.426\small{\pm0.011} & 0.217\small{\pm0.004} & 0.322\small{\pm0.004} & \mathbf{0.181\small{\pm0.003}} & \mathbf{0.283\small{\pm0.003}}  \newline
> \       & 336 & 0.247\small{\pm0.011} & 0.351\small{\pm0.009} & 0.241\small{\pm0.016} & 0.335\small{\pm0.013} & \mathbf{0.223\small{\pm0.019}} & \mathbf{0.318\small{\pm0.014}} & 0.369\small{\pm0.011} & 0.448\small{\pm0.009} & 0.236\small{\pm0.004} & 0.339\small{\pm0.003} & \mathbf{0.194\small{\pm0.002}} & \mathbf{0.297\small{\pm0.003}}  \newline
> \       & 720 & 0.270\small{\pm0.006} & 0.371\small{\pm0.003} & 0.259\small{\pm0.006} & 0.349\small{\pm0.003} & \mathbf{0.251\small{\pm0.021}} & \mathbf{0.342\small{\pm0.017}} & 0.396\small{\pm0.009} & 0.457\small{\pm0.005} & 0.267\small{\pm0.001} & 0.363\small{\pm0.001} & \mathbf{0.217\small{\pm0.001}} & \mathbf{0.317\small{\pm0.000}}  \newline
> \hline
> \hline
> \end{array}

---

> ### Comment · Reviewer_2uZk · 2022-08-06
> **Thank you for the responses**
>
> Thank you for the responses. The additional empirical results clarified many concerns that I have earlier, and I have raised the score accordingly.
>
> At the same time, from the added ablation study on EMA, we can see that EMA actually plays a more important role than the flooding technique emphasized in this paper. Therefore, it is essential to at least reposition the paper to emphasize this part instead of putting it as an ablation study in the Appendix.

---

> > ### Author Response · Authors · 2022-08-08
> > **Thank you for the update**
> >
> > Thank you for carefully reviewing our response!
> >
> > Following your suggestion, we will clarify the importance of EMA and move the ablation study on EMA to our main text.

---

### Official Review · Reviewer_dPWJ · 2022-07-14

**Rating:** 6
**Confidence:** 4
**Soundness:** 4 excellent
**Presentation:** 3 good
**Contribution:** 3 good

**Summary:**

The paper proposes a method to regularize the training of time-series forecasting models for better generalization performance. It extends the recently proposed flooding method which prevents the training loss from dropping to zero by regularizing it to stay above a fixed threshold. The authors argue that such fixed threshold is not amenable to time-series forecasting as it regularizes the average of all future time-steps and features. To address this issue, WaveBound is proposed which dynamically enforces the threshold for each time-step and feature per mini-batch. This is done by keeping an additional copy (called target) of the forecaster whose weights are updated with exponential moving average (EMA) of the source network. The source network is then regularized to be in the epsilon neighborhood of the loss predicted by the target network, thus dynamically regularizing the source network. Experiments have been presented on several datasets showing the improvement WaveBound-regularization offers for multiple forecasting models.

**Questions:**

- How was alpha, epsilon chosen?
- Do the authors have any comments on the applicability of the method to loss functions other than the MSE loss?

**Limitations:**

Limitations have been discussed by the authors in the appendix. While the method does introduce additional training overhead, there is no overhead at test time and the method improves the generalization performance significantly.

**Strengths And Weaknesses:**

[Strengths]

- This paper presents a novel extension of the previously proposed flooding regularization to time-series forecasting models. I like the innovative idea of using a slow network to dynamically set the flooding threshold.
- The proposed idea is technically sound. Key claims of the paper have been supported by systematic experiments. Several benchmark models have been studied on several datasets showing that WaveBound-regularization improves the generalization performance across these models. Qualitative results also solidify authors’ claims. Ablation experiments have also been conducted to highlight the importance of dynamically regularizing the model.
- The paper is well-written and easy to understand. I have some suggestions that could improve the readability and make the main text more complete.
    - Consider moving the pseudocode to the main text.
    - Include a proof-sketch for Theorem 1. In the current version, the theorem appears abruptly and isn’t followed-up with any discussion. Please consider adding discussion around the theorem to stress its significance.
- With its wide applicability, the proposed method and results are significant for the time-series community.

[Weaknesses]

- Discussion of some important prior work is completely missing from the paper. In section 5, the authors discuss statistical approaches and deep-learning-based approaches but complete miss on the important class of state-space models (and their combination with neural networks). A non-exhaustive list of recent works combining SSMs with NNs for time-series forecasting follows.
    - Kurle et al. “Deep Rao-blackwellised particle filters for time series forecasting.” In NeurIPS, 2020.
    - Rangapuram et al. “Deep state space models for time series forecasting.” NeurIPS, 2018.
    - de Bézenac et al. ”Normalizing kalman filters for multivariate time series analysis.” In NeurIPS, 2020.
    - Ansari et al. "Deep Explicit Duration Switching Models for Time Series." *In NeurIPS, 2021*.
- The proposed method is focused on a particular type (MSE) of loss function. Is the method applicable to other types of loss functions, e.g., those for generative models listed above?
- The method introduces additional training overhead as it requires another copy of the network. This weakness has also been acknowledged by the authors.

---

> ### Author Response · Authors · 2022-08-02
> **Response to Reviewer dPWJ [2/2]**
>
> **[C3, Q2] The applicability of our WaveBound to other types of loss functions.**
>
> Thank you for your constructive suggestions. Since the recent state-of-the-art time series forecasting methods, including Transformer-based methods (e.g., Autoformer), adopt the mean-squared error (MSE) as their objective functions by default, we mainly focus on using MSE in our paper. However, we remark that our approach can readily be applied to any losses which can be element-wisely computed with the same form of original loss in principle. For example, we can apply our proposed WaveBound with the MAE loss by letting $\hat{R}  _{jk}(g)=\frac{1}{N} \sum _{i=1}^N |g _{jk}(x_i)-(y_i) _{jk}|$
>
> To demonstrate the applicability of our WaveBound to a loss function other than MSE, we apply our WaveBound on a widely-used generative model, DeepAR[4], which is trained by minimizing the negative log-likelihood (NLL). As shown in the table below, the performance of DeepAR increases in terms of both NLL and MSE when we apply WaveBound to the model. In addition, we are working on generative models you listed up to verify that our WaveBound can also improve those models. We will add experimental results for these models in the final version of our paper.
>
> \begin{array}{lc| cc|cc}
> \hline
>  & & \rlap{\text{Origin}} &&\rlap{\text{w/ WaveBound}} \newline
> \hline
>  Metric & & NLL & MSE & NLL & MSE \newline
> \hline
> \ ECL   & 8   & 1.059\small{\pm0.152} & 0.447\small{\pm0.168} & \mathbf{0.935\small{\pm0.045}} & \mathbf{0.306\small{\pm0.001}} &  \newline
> \       & 16  & 1.168\small{\pm0.101} & 0.500\small{\pm0.208} & \mathbf{1.148\small{\pm0.052}} & \mathbf{0.371\small{\pm0.036}} &  \newline
> \       & 24  & 1.256\small{\pm0.073} & 0.617\small{\pm0.119} & \mathbf{1.180\small{\pm0.062}} & \mathbf{0.394\small{\pm0.043}} &  \newline
> \hline
> \hline
> \end{array}
>
> ---
>
> **[Q1] How were $\alpha$ and $\epsilon$ chosen?**
>
> The hyperparameter $\alpha$ is used to control the update speed of the EMA model, i.e., the target network. In various approaches that utilize EMA technique [5, 6], $\alpha$ is mostly searched in {0.99, 0.999, 0.9999}. In all experiments, we use $\alpha=0.99$ by default since the update speed of the target network becomes too slow if we set it greater than 0.99.
>
> The hyperparameter $\epsilon$ indicates how far the error bound of the source network can be from the error of the target network. For simplicity, we searched $\epsilon$ only in {0.01, 0.001} and selected the best $\epsilon$ for each dataset. To empirically show the effects of $\epsilon$, we also examine the results of choosing different $\epsilon$ in Appendix E.
>
> ---
>
> [1] Rangapuram et al. “Deep State Space Models for Time Series Forecasting.”, NeurIPS, 2018.\
> [2] Krishnan et al. “Structured Inference Networks for Nonlinear State Space Models.”, AAAI, 2017.\
> [3] Klushyn et al. “Latent Matters: Learning Deep State-Space Models.”, NeurIPS, 2021.\
> [4] Flunkert et al. “DeepAR: Probabilistic Forecasting with Autoregressive Recurrent Networks.”, CoRR abs/1704.04110, 2017.\
> [5] Grill et al. “Bootstrap Your Own Latent - A New Approach to Self-Supervised Learning.”, NeurIPS, 2020.\
> [6] He et al. “Momentum Contrast for Unsupervised Visual Representation Learning.”, CVPR, 2020.

---

> ### Author Response · Authors · 2022-08-02
> **Response to Reviewer dPWJ [1/2]**
>
> **[C1] Editorial comments for improving readability.**
>
> We appreciate your constructive comments on improving the readability of our paper. First, you suggested moving the section of pseudo-code to our main paper from Appendix. Due to the limited space of the paper, we decided to add instructions to the pseudo-code in the Method section to guide the user to easily access the pseudo-code. We will reconsider introducing the pseudo-code in our main paper in the camera-ready version.
>
> You also suggested that explanations (proof-sketch) and discussion would be needed for Theorem 1. In the revised version, we added an intuitive explanation for Theorem 1 and faithfully described the connection between Theorem 1 and our main motivation in the Method section. Thank you again for your precious recommendations!
>
> ---
>
> **[C2] Including state-space models as the related works.**
>
> As you mentioned, state space models (SSMs) have been well studied for time series forecasting. Moreover, several studies attempted to fuse the advantages of SSMs and that of neural networks [1, 2, 3]. Following your suggestion, we updated our Related Works section to include the state-space models along with the statistical approaches and deep-learning-based approaches.

---

> ### Comment · Reviewer_dPWJ · 2022-08-05
> **Post Rebuttal Comment**
>
> Thank you to the authors for their responses and additional empirical results. Having read the other reviews and authors' responses, my overall opinion about the paper remains unchanged — this is a good paper with significant results for the time series community.

---

### Author Response · Authors · 2022-08-02
**General Response**

Dear reviewers and AC,

We sincerely appreciate your efforts in reviewing our manuscripts: WaveBound: Dynamically Bounding Error for Stable Time Series Forecasting.

We are grateful that reviewers found that our main idea is innovative (dPWJ) and technically sound (dPWJ), and our proposed WaveBound is novel (dPWJ, v7br) and shows its effectiveness in extensive experiments (dPWJ, 2uZk, v7br, TFyo). As the reviewers mentioned, we hope that our WaveBound will provide meaningful insights to the machine learning community by highlighting the importance of dynamically regularizing the model.

Within the short response period, we did our best to reflect all reviewers’ feedback. In addition to answering each reviewer’s questions, we also faithfully revised our manuscript following the suggestions. The revisions made during this period are highlighted in blue.

Sincerely,

Authors.

---

### Author Response · Authors · 2022-08-08
**Thank you for your thoughtful discussions**

Dear reviewers,

We greatly appreciate your efforts in reviewing our manuscript.

We hope that our responses and discussions have addressed the reviewers’ concerns.

In case of any further issues, we will do our best to address them as soon as possible.

Thank you very much again for your time and efforts in giving constructive reviews.

Sincerely,

Authors.

---

### Meta-Review · Area_Chair_Digj · 2022-08-27

**Recommendation:** Accept
**Confidence:** Less certain

**Metareview:**

The reviewers highlight the novelty of the method, the clarity of writing, and the consistent performance improvements over baselines. Initial concerns by the reviewers related to missing related work, missing empirical evaluation within the more mature short-term forecasting experimental paradigm, and missing empirical comparisons (comparing to reversible instance norm, isolation of the effect of EMA) were addressed by the authors during the discussion period. Some concerns around the overall motivation behind the proposed approach remain, but are outweighed by the empirical effectiveness.

**Award:**

No

---

### Decision · Program_Chairs · 2022-09-14

Accept